# Effect of Biofeedback Therapy during Temporary Stoma Period in Rectal Cancer Patients: A Prospective Randomized Trial

**DOI:** 10.3390/jcm10215172

**Published:** 2021-11-04

**Authors:** Hyeon-Min Cho, Hyungjin Kim, RiNa Yoo, Gun Kim, Bong-Hyeon Kye

**Affiliations:** 1Department of Surgery, St. Vincent’s Hospital, College of Medicine, The Catholic University of Korea, Suwon 16247, Korea; hmcho@catholic.ac.kr (H.-M.C.); ninayoo1111@gmail.com (R.Y.); vertex988304@naver.com (G.K.); 2Department of Surgery, Eunpyeong St. Mary’s Hospital, College of Medicine, The Catholic University of Korea, Seoul 06591, Korea; hj@catholic.ac.kr

**Keywords:** biofeedback therapy, defecation dysfunction, rectal cancer, sphincter-preserving surgery

## Abstract

Background: This prospective randomized controlled study was designed to evaluate the effect of biofeedback therapy (BFT) during temporary stoma period to prevent defecation dysfunction after sphincter-preserving surgery (SPS). Methods: Following SPS with temporary stoma, patients were divided according to whether (BFT group) or not (Control group) they received BFT. BFT was performed once or twice a week during the temporary stoma period. Kegel exercise were advised to all the patients. Subjective defecation symptoms were evaluated according to Cleveland Clinic Incontinence Score (CCIS) as primary outcome at 12 months postoperatively. Manometric data of five time-points were also analyzed. Results: Twenty-one patients in the BFT group and 23 patients in the control group received anorectal physiologic testing. The incidence of CCIS of more than 9 points, which is the primary end point in this study, was not statistically different between BFT group and control group (*p* = 1.000). The liquid stool incontinence in the BFT group showed a better tendency (*p* = 0.06) at 12 months post-SPS. Time-dependent serial changes in maximal sensory threshold (Max RST) was significantly different between the BFT and control groups (*p* = 0.048). Also, the change of mean resting pressure (MRP) tended to be more stable in the BFT group (*p* = 0.074). Conclusions: The BFT in the period of temporary stoma may be related to liquid stool incontinence at 12 months post-SPS and lead to stable MRP and better Max RST. Therefore, BFT during temporary stoma might be helpful for preventing and minimizing defecation dysfunction in high risk patients after SPS, NCT01661829).

## 1. Introduction

Recently, sphincter-preserving surgery (SPS), including low anterior resection (LAR), ultra-LAR, and intersphincteric resection (ISR), is frequently performed for the treatment of rectal cancer patients [1,2]. It has been previously reported that the oncologic outcomes of SPS are not inferior to those of Mile’s operation, even if the tumor is close to the anal verge [3,4]. SPS presumably translates into better quality of life compared to an abdominoperineal resection through the avoidance of a permanent stoma [5]. However, SPS may lead to significant bowel dysfunction, known as low anterior resection (LARS), which can adversely affect QoL [6]. Some reports demonstrated similar QoL in patients who underwent either SPS or Mile’s operation, primarily because QoL is partly attributed to the defecation problem arising from bowel dysfunction after surgery [7,8,9].

Currently, many colorectal surgeons are interested in defecation dysfunction after SPS in rectal cancer patients and some studies have focused on overcoming this defecation dysfunction [10,11,12,13]. Firstly, many colorectal surgeons are trying to design some studies to identify the risk factors related to LARS. Most of the studies demonstrated that the low-lying tumor, low level of anastomosis near the anal verge, radiation therapy, and diverting stoma might be the potential risk factors related to LARS [14]. However, there are no specific treatments for LARS and most of the treatment approaches are empirical and symptomatic, using usual treatment options for defecation dysfunction, including loperamide, anal plugs, biofeedback therapy (BFT), rectal irrigation, and sacral or tibial neuromodulation [11,15].

Among these approaches, BFT is frequently recommended for patients with fecal incontinence and are not responsive to medical therapy. BFT is safe, non-invasive, and inexpensive, with practically few adverse effects [16,17]. Hence, because the majority of LARS patients exhibit defecation symptoms related to fecal incontinence or urgency, BFT might be an effective therapeutic strategy for LARS patients [18].

Till now, these treatment strategies are empirical and employed after diagnosis of LARS, despite its relatively high incidence after SPS. However, lots of studies on LARS have dealt with treatment after the occurrence of LARS, and there are few studies on the prevention of LARS or the timing of the therapeutic approaches for LARS.

We conducted a prospective randomized controlled study to evaluate the effect of BFT in the period of a temporary stoma on the defecation function in patients who underwent neoadjuvant chemoradiation therapy (nCRT) followed by SPS. In 2016, the interim report of this randomized controlled trial was published, demonstrating that BFT might be meaningful in maintaining resting anal sphincter tone without a preventive effect of defecation dysfunction at 6 months after SPS [13]. In this study, the data were analyzed to evaluate the effect of BFT during temporary stoma on the defecation function at about 1 year after SPS.

## 2. Materials and Methods

This study is registered on clinicaltrials.gov (NCT01661829) (accessed on 8 October 2012).

### 2.1. Ethics

After obtaining the review board approval from St. Vincent’s Hospital, The Catholic University of Korea, CMC Clinical Research Coordination Center (VC12EISI0023), patients were enrolled in the study and their clinical information was prospectively collected.

### 2.2. Eligibility Criteria

#### 2.2.1. Inclusion Criteria

For inclusion in this study, patients must fulfill the following requirements pre-operatively: (1) pathologically proven adenocarcinoma; (2) primary tumor located in the rectum, 12 cm below from anal verge; (3) well-maintained fecal continence before nCRT; (4) long-course nCRT (1.8 Gy/day, five fractions per week, and a total dose of 50.4 Gy/28 fractions + two cycles of concurrent chemotherapy with radiotherapy (5-fluorouracil (5-FU), 400 mg/m^2^ (i.v.), 1 h before radiotherapy and leucovorin, 20 mg/m^2^ (i.v.), immediately before each dose of 5-FU on days 1–5 and days 29–33); (5) temporary stoma during SPS at 6-10 weeks after nCRT; (6) adequate organ functions; and (7) written informed consent.

#### 2.2.2. Exclusion Criteria

The exclusion criteria included: (1) fecal incontinence prior to nCRT; (2) stoma prior to SPS due to obstructive lesion, bleeding, fistula, etc.; (3) active infectious disease requiring systemic therapy; and (4) pregnant women.

### 2.3. Randomization and Sample Size

After nCRT following SPS with temporary stoma, patients were randomized according to whether they received BFT (control group) and were assigned 1:1 with or without BFT. Using a random-number table with assignment codes concealed in opaque envelopes, half of the patients were randomized to the BFT group and the other half to control group. Signed informed consent was obtained from all the patients prior to randomization. A two-sided test with a significance level of 0.05 and power of 80% was used to evaluate the sample size requirements. We estimated that a sample size of 56 patients (28 patients each in the BFT and control group) would be required to detect a 35% reduction in the incidence of CCIS of more than 9 point at 1 year after SPS.

### 2.4. Biofeedback Therapy & Kegel Exercise

The biofeedback therapy designed to strengthen the muscles of the external anal sphincter was performed by recording the strength of the perineum and abdominal muscles, combined with a visual/audible signals proportional to the pressures themselves. In this study, any probe which is inserted into anal canal was not used to avoid any injury of anastomosis. The patient was instructed to slowly contract and relax the external anal sphincter while sitting in a chair with the sensor touching directly around the anus. Biofeedback techniques provide the patient with information about anal sphincter pressure or activity related to the patient’s performance. The patient was encouraged to relax abdominal muscles while maximally squeezing the muscles surrounding the anal canal. This maneuver was performed repeatedly. While receiving visual feedback and verbal instructions on how to reach this goal, the patient had to squeeze to prevent defecation. The patient can also be taught to suppress false responses, such as contractions of the abdominal muscles.

All patients were encouraged to take a Kegel exercise after SPS, even if the patient had a temporary stoma. The education program for Kegel exercise was performed by specialized nurse who is a wound, ostomy, and continence nurse (WOCN). The patients were taught to perform Kegel exercise by starting a small number of exercises in a short amount of time, gradually increasing both the length and number of exercises, by lifting and holding the pelvic floor muscles and then relaxing them. At least two sets of the exercise a day was our recommendation.

### 2.5. Study Design

From March 2012 to February 2014, a total of 56 patients who underwent nCRT following SPS with temporary stoma were enrolled in our study. BFT was performed one or twice a week during temporary stoma period. All the patients were advised to undergo conservative self-rehabilitations, such as Kegel exercises. To evaluate the anorectal function, anorectal manometry, transanal ultrasound, and Cleveland Clinic Incontinence Score (CCIS) were performed at the following time points: before nCRT (Period 1), 4 weeks after the completion of nCRT (Period 2), before the reversal of temporary stoma (Period 3), 6 months after SPS with temporary stoma (Period 4), and 12 months after SPS with temporary stoma (Period 5). Patients were randomly assigned to one of two groups just before first adjuvant chemotherapy after SPS with temporary stoma. (Figure 1) Subjective defecation symptoms were evaluated as CCIS, mean daily defecation frequency, severity of incontinence (none, urgency to evacuate, soiling, and accidents), and requirement of antidiarrheal drugs for each period. We also estimated the treatment response with objective parameters using anorectal manometry. The “degree of change” was considered as the rate of change of the manometric data based on the data from Period 1 (manometric data in each Period/manometric data in Period 1). In addition, the response was measured as the mean of the all “degrees of change” at each Period (=“measure of response”). In this study, the “degree of change” and the “measure of response” were used as the main comparative parameters between the BFT and control groups.

### 2.6. Primary End Point

The primary end point of our study was to identify differences in the incidence of bowel dysfunction, particularly severity of fecal incontinence, between BFT and control groups. The defecation dysfunction was defined as CCIS of 9 points or higher.

### 2.7. Secondary End Points

In addition, the “measure of response” according to the initial tumor location and method of anastomosis was compared. Patients were divided into two groups according to whether the lower margin of the tumor observed to be >5 cm or ≤5 cm from the anal verge using rigid proctosigmoidoscopy. For patients with tumors <5 cm from the anal verge, an additional boost field to the surrounding area included the anal canal. On the other hand, for patients with tumors >5 cm from the anal verge, the anal canal and perineum were excluded from an additional boost areas. Stapled colorectal anastomosis (S-CR) was performed using the double stapling technique via an intra-abdominal approach. Hand-sewn coloanal anastomosis (H-CA) was performed using the hand-sewn technique through a perianal approach.

### 2.8. Statistical Analyses

Continuous variables were compared using the Student’s *t*-test and one-way ANOVA and expressed as mean ± SD. Categorical variables were analyzed using the χ2 test. The various measurements for “degree of change” and “measure of response” for each Period were compared using a generalized linear models. Significance was defined as a *p* ≤ 0.05. All statistical analyses were performed using the Statistical Package of the Social Sciences (SPSS) version 15.0 for Windows (SPSS, Inc., Chicago, IL, USA).

## 3. Results

Figure 2 shows the flow of participants through the recruitment and randomization phases of the study. A total of 87 patients were screened for eligibility. Of these, 14 patients did not meet the inclusion criteria, 17 patients refused to participate in this study, seven patients gave up physiological examinations every period due to economic reasons, five patients lived far from our hospital, and five patients refused enrollment in this study for unknown causes. Therefore, we randomized 56 patients with baseline measurements and informed consent prior to randomization to the intervention conditions. In the BFT group, one patient failed follow-up, three patients withdrew consent during the study period, one patient suffered an anastomosis leak after LAR, and two patients could not undergo physiological testing due to postoperative anal strictures. In the control group, three patients failed follow-up and two patients withdrew consent during the study period. Finally, 21 patients in the BFT group and 23 patients in the control group underwent anorectal physiologic testing by Period 5.

The demographic and baseline characteristics of the participants are presented in Table 1. BFT and control participants did not differ significantly based on demographic or baseline characteristic.

Table 2 shows the defecation symptoms of patients in Period 5. There was no statistically significant difference between the BFT group and the control group in the incidence of CCIS with a score of 9 or higher, the primary endpoint of this study (*p* = 1.000). Although there was no statistical significance, the results in the BFT group showed better tendencies in terms of liquid stool incontinence (*p* = 0.06). Initial tumor location was significantly related to lifestyle alteration (*p* = 0.011). Also, anastomosis location was significantly related to gas incontinence (*p* = 0.017). Since the patients entered the anal defecation phase after Period 3, the serial changes in CCIS during Period 4 and Period 5 were analyzed (Figure 3). Although not statistically significant, S-CR anastomosis, higher location of anastomosis, and BFT were correlated with lower CCIS in Period 5 compared to Period 4.

Table 3 shows the “degree of change” for each manometric parameter in period 5. There were no significant differences in the “measure of response” for any manometric parameter in Period 5 based on treatment strategy, initial tumor location, anastomosis location, or anastomosis method.

The time-dependent changes in the “measure of response” for mean resting pressure (MRP), maximal squeeze pressure (MSP), maximal rectal sensory threshold (Max RST), and rectal compliance (RC), are shown in Figure 4 and Figure 5 (according to the CCIS during Period 5 and treatment options, respectively).

There were no significant “measure of response” for manometric parameter related to the CCIS in Period 5. Although not statistically significant, “measure of response” for maximal RST tended to improve in the patients with lower CCIS in Period 5 (*p* = 0.095) (Figure 4). The time-dependent ‘measure of response’ for Max RST were significantly different between the BFT and control groups (*p* = 0.048). Although there was no statistical significance, the “measure of response” for MRP tended to be more stable in the BFT group (*p* = 0.074). However, there were no significant differences between groups in the time-dependent changes in the “measure of response” for MSP and RC (Figure 5).

## 4. Discussion

SPS provides an opportunity to prevent changes in the patients’ body structure that are important to the physical and emotional well-being of patients with rectal cancer [19,20]. However, after SPS, about 60–90% of patients develop a change in the bowel habits, including fecal incontinence, and urgent and frequent bowel movements, known as LARS [14]. Unfortunately, despite high incidence of defecation dysfunction after SPS, most treatments for LARS are applied after the emergence of symptoms. In addition, the etiology and pathophysiology of LARS are not well understood. Radiation therapy, tumor location, the location of anastomosis, the extent of operation (lateral pelvic lymph node dissection), post-operative chemotherapy, mechanical bowel preparation, and temporary stoma are risk factors for LARS [14,21]. All patients enrolled in this study received nCRT followed by TME, mechanical bowel preparation before radical surgery, planned temporary stoma during radical surgery, and post-operative adjuvant chemotherapy. Therefore, all the patients in our study had very high risk for developing LARS.

Generally, most surgeons may recommend pelvic muscle rehabilitation, such as Kegel exercises after SPS. This type of exercise may be beneficial in cases of fecal incontinence and pelvic organ prolapse [22]. However, the correct execution of these exercises is not monitored by medical staff, making it difficult to determine whether the training was ineffective owing to inherent inefficiency, or because it was incorrectly performed [23,24]. On the other hand, BFT can provide visual information about the activity of the patient’s pelvic floor muscles. Hence, BFT can indicate improvement in patient’s pelvic floor muscle strength, which can also be monitored by medical staff [25]. This is a key difference between BFT and Kegel exercises. BFT is a noninvasive, inexpensive approach with minimal adverse effects [15]. BFT, in addition to Kegel exercises, might be a suitable and safe option for application during the anal resting phase with temporary stoma. The primary aim in this study was to lower the incidence of defecation dysfunction using BFT during temporary stoma period. Around 57.1% and 60.9% of patients in the BFT and control groups, respectively, had more than 9 points of CCIS (mean ± SD: 10.05 ± 5.2 vs. 10.17 ± 5.3; *p* = 1.000) (Table 2). We did not meet the primary end point successfully. However, with respect to “measure of response,” the change of maximal RST was significantly minimized in the BFT group (*p* = 0.048). In addition, the change of MRP tended to minimize in the BFT group though the change was not statistically significant (*p* = 0.074). (Figure 5) One study suggested that anorectal manometry has excellent sensitivity, moderate specificity, and convincing accuracy in assessing anorectal function [26]. Therefore, in the evaluation of fecal incontinence patients, the discriminatory power of anorectal manometry is sufficiently high to justify its clinical use. In the present study, anorectal manometry was applied to evaluate the mechanism of defecatory dysfunction. In general, RST was one of indicators for rectal hypersensitivity and MRP for the capability to close the anus at resting state to prevent fecal incontinence. Although recto-anal coordination, which is reflected by the rectoanal inhibitory reflex (RAIR), may be more important for recto-anal hypersensitivity [27], the combination of RST and MRP can be an alternative indicator for the evaluation of recto-anal hypersensitivity. Hence, our results indicated that BFT during temporary stoma period could minimize the changes in recto-anal hypersensitivity and internal anal sphincter function. Based on these results, the authors suggested that BFT during temporary stoma period can be recommended to patients with high risk for LARS.

Liquid stool emptied rapidly into the rectum results in great stress on the sphincters, and, even in normal subjects, phasic flows of liquid stool may occasionally produce urgency and incontinence [28]. Also, because rapid colonic transit time in left sided colon after SPS from the denervation during the operation, liquid stool may intensify defecation dysfunction after SPS [29]. Therefore, liquid stool incontinence may be one of most important symptoms related to postoperative defecation dysfunction. In present study, Patients with low level anastomosis above 3 cm from anal verge (*p* = 0.051) and those who underwent BFT during stoma period (*p* = 0.06) lesser frequently complained liquid stool incontinence. (Table 2) Anastomosis location is hard to modify to control defecation dysfunction after SPS, but BFT is relatively easy and feasible to apply to the patient who undergo SPS. Although there was not statistically significant, BFT during stoma period may be a part of treatment strategy to prevent or minimize postoperative defecation dysfunction by decreasing liquid stool incontinence.

There is little recommendation about a combination or planned serial application of various treatment options from the beginning of treatment. During a temporary stoma period, it is not always feasible to use an invasive approach, including sacral nerve stimulation (SNS), probiotics to the colon, or transanal irrigation. However, BFT or medication might be more feasible and safer options during this period. Currently, SNS is reported as the best treatment modality for LARS, subject to availability [15]. Hence, a serial treatment approach, including preventive BFT followed by SNS, could be employed to improve the outcome for LARS. A French study demonstrated that the stool consistency and low stimulation intensity were predictive factors for a successful outcome of SNS [30]. According to the analysis of the manometric data in this study, a higher MRP before SNS was related to success at the 6-month follow-up after permanent implantation. Taken together, by improving rectal and anal sensitivity, BFT during temporary stoma may be a potentially useful treatment strategy for LARS and for eliciting improved responses for future SNS.

The limitations of our study are that this was designed and conducted in a single center, a small number of patients were included, and the drop-out rate (21.4%) was somewhat high. Also, in our study, sufficiently detailed instructions for Kegel exercises could not be provided. In addition, anti-diarrheal drugs were prescribed without any criterion. However, there are not any reports for improving LARS with treatment modality during anal resting phase. Even though BFT during the temporary stoma period failed to improve the symptomatic scores, we found that BFT, during this period, might be effective for the maintenance of rectal and anal hypersensitivity. Moreover, BFT during temporary stoma may provide better conditions for the application of additional treatments, including SNS.

## 5. Conclusions

The BFT during temporary stoma period did not improve the patient’s symptom scores. However, it was helpful in maintaining the recto-anal sensitivity and internal anal sphincter status and minimizing the liquid stool incontinence. It is necessary to design an appropriate treatment during anal resting phase for patients with high risk of LARS. In the future, a large-scale randomized trial is expected to draw more reliable results.

## Figures and Tables

**Figure 1 jcm-10-05172-f001:**
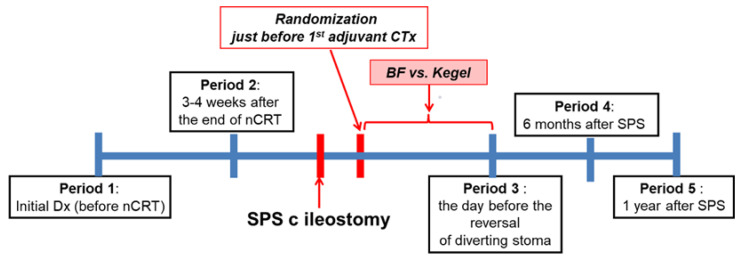
Timeline of treatment and evaluation in this study. SPS, sphincter-preserving surgery; CRT, chemoradiation therapy; CTx, chemotherapy; BF, biofeedback.

**Figure 2 jcm-10-05172-f002:**
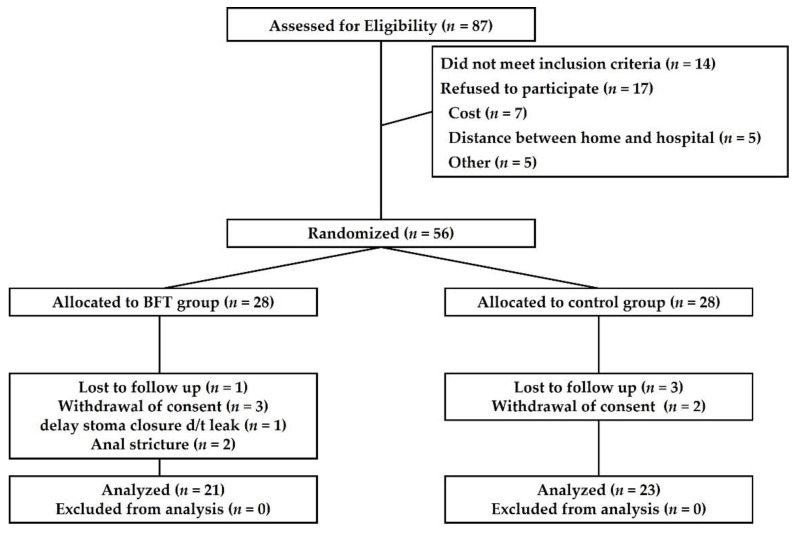
Flowchart of the inclusion process. BFT, biofeedback therapy.

**Figure 3 jcm-10-05172-f003:**
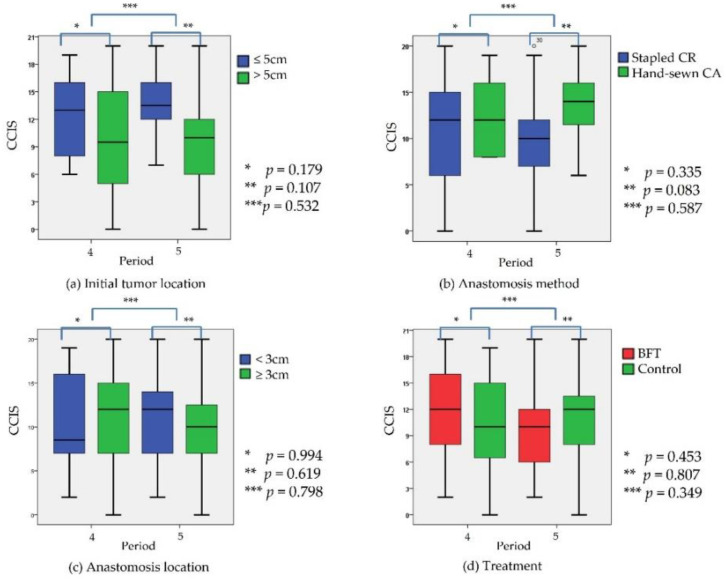
Changes in patient’s symptom scores (Cleveland Clinic Incontinence Score, CCIS) between Period 4 (6 months after radical surgery) and Period 5 (1 year after radical surgery). There was no significant difference in the change of CCIS between Period 4 and Period 5 according to initial tumor location (**a**), anastomosis method (stapled colorectal vs. hand-sewn coloanal) (**b**), the distance of anastomosis from anal verge (**c**), and the treatment option during temporary stoma period (**d**). Lower CCIS during Period 5 correlated with stapled colorectal anastomosis, higher location of anastomosis, and biofeedback therapy (BFT); however, the correlation was not statistically significant.

**Figure 4 jcm-10-05172-f004:**
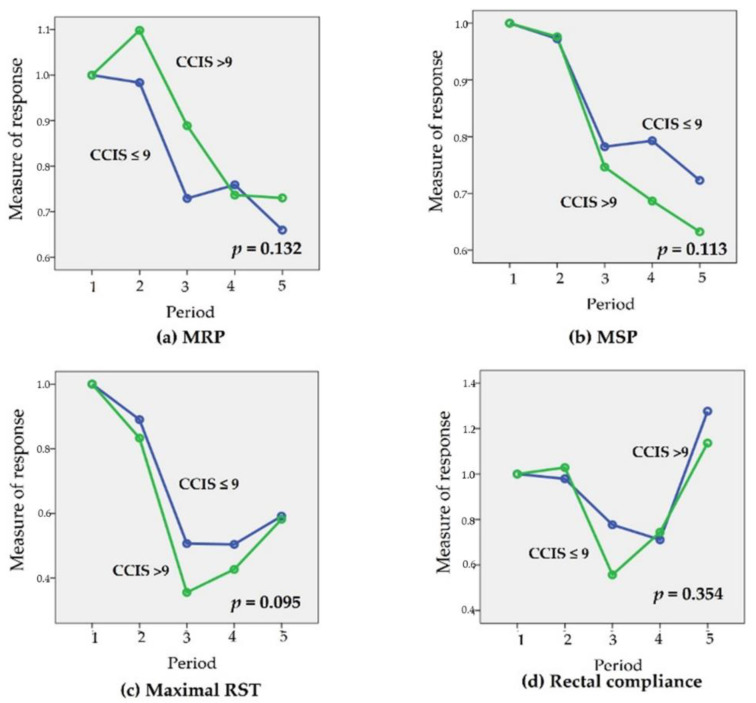
Changes in the ‘measure of response’; (**a**) for mean resting pressure (MRP), (**b**) for maximal squeezing pressure (MSP), (**c**) for maximal rectal sensory threshold (RST), and (**d**) for rectal compliance according to the 9 points of CCIS in Period 5. There was no significant difference in the change in the ‘measure of response’ for MRP, MSP, maximal RST, and rectal compliance according to 9 points of CCIS in Period 5. Although not statistically significant, “measure of response” for maximal RST tended to improve in the patients with lower CCIS in Period 5 (*p* = 0.095).

**Figure 5 jcm-10-05172-f005:**
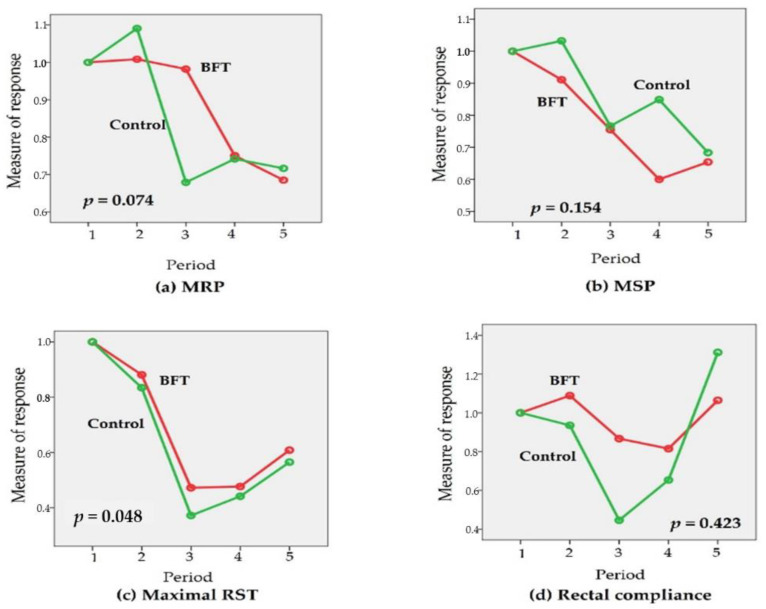
Changes in the ‘measure of response’; (**a**) for mean resting pressure (MRP), (**b**) for maximal squeezing pressure (MSP), (**c**) for maximal rectal sensory threshold (RST), and (**d**) for rectal compliance according to the treatment during temporary stoma period. There was significant time-dependent changes in ‘measure of response’ for maximal RST between the BFT and control groups (*p* = 0.048). Although there was no statistical significance, the “measure of response” for MRP tended to be more stable in the BFT group (*p* = 0.074).

**Table 1 jcm-10-05172-t001:** Demographic data of patients enrolled into this study.

		BFT Group (*n* = 21)	Control Group (*n* = 23)	*p*-Value
Age	≤65 years	15 (71.4%)	12 (52.2%)	
	>65 years	6 (28.6%)	11 (47.8%)	0.228
Sex	male	11 (52.4%)	14 (60.9%)	
	female	10 (47.6%)	9 (39.1%)	0.761
Initial tumor location (cm from anal verge)	≤5 cm	5 (23.8%)	5 (21.7%)	
	>5 cm	16 (76.2%)	18 (78.3%)	1.000
	Mean ± SD	7.26 ± 2.07	7.83 ± 2.59	0.433
Anastomosis method	S-CR	16 (76.2%)	21 (91.3%)	
	H-CA	5 (23.8%)	2 (8.7%)	0.232
Location of anastomosis (cm from anal verge)	<3 cm	6 (28.6%)	3 (13.0%)	
	≥3 cm	15 (71.4%)	20 (87.0%)	0.272
	Mean ± SD	3.71 ± 1.48	3.96 ± 1.46	0.589

BFT, biofeedback therapy; S-CR, Stapled colorectal; H-CA, Hand-sewn coloanal.

**Table 2 jcm-10-05172-t002:** Defecation symptoms at Period 5 which means ‘1 year after sphincter-preserving surgery’.

			Initial Tumor Location (From Anal Verge)			Anastomosis Method			Anastomosis Location (From Anal Verge)			Treatment		
			≤5 cm	>5 cm	*p*-Value	S-CR	H-CA	*p*-Value	< 3cm	≥3 cm	*p*-Value	BFT Group	Control Group	*p*-Value
CCIS	≤9		2 (20.0%)	16 (47.1%)		17 (45.9%)	1 (14.3%)		3 (33.3%)	15 (42.9%)		9 (42.9%)	9 (39.1%)	
	>9		8 (80.0%)	18 (52.9%)	0.161	20 (54.1%)	6 (85.7%)	0.211	6 (66.7%)	20 (57.1%)	0.716	12 (57.1%)	14 (60.9%)	1.000
Type of incontinence	Gas	Never	0	7 (20.6%)	0.066	7 (18.9%)	0	0.245	1 (5.2%)	6 (24%)	0.017	4 (19.0%)	3 (13.0%)	0.071
		Rarely	0	5 (14.7%)		4 (10.8%)	1 (14.3%)		1 (5.2%)	4 (16%)		4 (19.0%)	1 (4.3%)	
		Sometimes	1 (10.0%)	7 (20.6%)		8 (21.6%)	0		1 (5.2%)	7 (28%)		1 (4.8%)	7 (30.4%)	
		Usually	6 (60.0%)	13 (38.2%)		15 (40.5%)	4 (57.1%)		13 (68.4%)	6 (24%)		11 (52.4%)	8 (34.8%)	
		Always	3 (30.0%)	2 (5.9%)		3 (8.2%)	2 (28.6%)		3 (16.0%)	2 (8%)		1 (4.8%)	4 (17.4%)	
	Liquid	Never	1 (10.0%)	6 (17.6%)	0.368	7 (18.9%)	0	0.372	1 (5.3%)	6 (24.0%)	0.051	1 (4.8%)	6 (26.1%)	0.06
		Rarely	0	4 (11.8%)		4 (10.8%)	0		0	4 (16.0%)		4 (19.0%)	0	
		Sometimes	1 (10.0%)	9 (26.5%)		9 (24.3%)	1 (14.3%)		4 (21.1%)	6 (24.0%)		6 (28.6%)	4 (17.4%)	
		Usually	7 (70.0%)	13 (38.2%)		15 (40.5%)	5 (71.4%)		13 (68.4%)	7 (28.0%)		8 (38.1%)	12 (52.2%)	
		Always	1 (10.0%)	2 (5.9%)		2 (5.4%)	1 (14.3%)		1 (5.3%)	2 (8.0%)		2 (9.5%)	1 (4.3%)	
	Solid	Never	1 (10.0%)	6 (17.6%)	0.851	7 (18.9%)	0	0.752	2 (10.5%)	5 (20.0%)	0.542	3 (14.3%)	4 (17.4%)	0.276
		Rarely	1 (10.0%)	6 (17.6%)		6 (16.2%)	1 (14.3%)		2 (10.5%)	5 (20.0%)		6 (28.6%)	1 (4.3%)	
		Sometimes	4 (40.0%)	8 (23.5%)		10 (27.0%)	2 (28.6%)		6 (31.6%)	6 (24.0%)		5 (23.8%)	7 (30.4%)	
		Usually	3 (30.0%)	11 (32.4%)		11 (29.7%)	3 (42.9%)		8 (42.1%)	6 (24.0%)		5 (23.8%)	9 (39.1%)	
		Always	1 (10.0%)	3 (8.8%)		3 (8.1%)	1 (14.3%)		1 (5.3%)	3 (12.0%)		2 (9.5%)	2 (8.7%)	
	Use of pad	Never	4 (40.0%)	19 (55.9%)	0.321	20 (54.1%)	3 (42.9%)	0.588	10 (52.6%)	13 (52.0%)	0.553	10 (47.6%)	13 (56.5%)	0.790
		Rarely	0	1 (2.9%)		1 (2.7%)	0		0	1 (4.0%)		0	1 (4.3%)	
		Sometimes	0	5 (14.7%)		5 (13.5%)	0		1 (5.3%)	4 (16.0%)		3 (14.3%)	2 (8.7%)	
		Usually	2 (20.0%)	3 (8.8%)		4 (10.8%)	1 (14.3%)		2 (10.5%)	3 (12.0%)		3 (14.3%)	2 (8.7%)	
		Always	4 (40.0%)	6 (17.6%)		7 (18.9%)	3 (42.9%)		6 (31.6%)	4 (16.0%)		5 (23.8%)	5 (21.7%)	
	Lifestyle alteration	Never	0	7 (20.6%)	0.011	7 (18.9%)	0	0.426	2 (10.5%)	5 (20.0%)	0.195	2 (9.5%)	5 (21.7%)	0.096
		Rarely	0	5 (14.7%)		4 (10.8%)	1 (14.3%)		2 (10.5%)	3 (12.0%)		5 (23.8%)	0	
		Sometimes	0	10 (29.4%)		9 (24.3%)	1 (14.3%)		2 (10.5%)	8 (32.0%)		5 (23.8%)	5 (21.7%)	
		Usually	8 (80.0%)	9 (26.5%)		14 (37.8%)	3 (42.9%)		11 (57.9%)	6 (24.0%)		6 (28.6%)	11 (47.8%)	
		Always	2 (20.0%)	3 (8.8%)		3 (8.1%)	2 (28.6%)		2 (10.5%)	3 (12.0%)		3 (14.3%)	2 (8.7%)	
Severity of incontinence	no		0	5 (14.7%)		5 (13.5%)	0		0	5 (14.3%)		1 (4.8%)	4 (17.4%)	
	urgency to evacuate		7 (70.0%)	23 (67.6%)		25 (67.6%)	5 (71.4%)		8 (88.9%)	22 (62.9%)		16 (76.2%)	14 (60.9%)	
	soling		1 (10.0%)	6 (17.6%)		6 (16.2%)	1 (14.3%)		0	7 (20.0%)		3 (14.3%)	4 (17.4%)	
	accidents		2 (20.0%)	0	0.036	1 (2.7%)	1 (14.3%)	0.439	1 (11.1%)	1 (2.9%)	0.182	1 (4.8%)	1 (4.3%)	0.575
Use of loperamide	no		0	11 (32.4%)		10 (27.0%)	1 (14.3%)		2 (22.2%)	9 (25.7%)		7 (33.3%)	4 (17.4%)	
	yes		10 (100%)	23 (67.6%)	0.046	27 (73.0%)	6 (85.7%)	0.659	7 (77.8%)	26 (74.3%)	1.000	14 (66.7%)	19 (82.6%)	0.303
No of daily defecation			8.6 ± 4.4	6.1 ± 4.3	0.119	6.5 ± 4.6	7.1 ± 2.7	0.767	8.4 ± 5.1	6.2 ± 4.1	0.182	7.71 ± 4.9	5.74 ± 3.7	0.145

CCIS, Clevelenad Clinic Incontinence Score; S-CR, stapled colorectal; H-CA, hand-sewn coloanal.

**Table 3 jcm-10-05172-t003:** The ‘degree of change’ according to manometric parameters at Period 5.

	Initial Tumor Location (from Anal Verge)			Anastomosis Method			Anastomosis Location (from Anal Verge)			Treatment		
‘Degree of Change’ at Peirod 5	≤5cm	>5cm	*p*-Value	S-CR	H-CA	*p*-Value	<3cm	≥3cm	*p*-Value	BFT Group	Control Group	*p*-Value
MRP	0.54 ± 0.49	0.75 ± 0.34	0.137	0.71 ± 0.40	0.67 ± 0.30	0.806	0.76 ± 0.49	0.69 ± 0.36	0.606	0.69 ± 0.35	0.72 ± 0.42	0.793
MSP	0.48 ± 0.41	0.72 ± 0.34	0.073	0.68 ± 0.38	0.61 ± 0.32	0.662	0.63 ± 0.34	0.68 ± 0.38	0.715	0.65 ± 0.31	0.68 ± 0.43	0.797
R asymmetry	1.22 ± 1.60	1.02 ± 0.38	0.710	1.06 ± 0.86	1.10 ± 0.53	0.907	1.56 ± 1.59	0.94 ± 0.39	0.278	1.03 ± 0.46	1.10 ± 1.04	0.768
S asymmetry	0.91 ± 0.65	1.20 ± 0.46	0.125	1.11 ± 0.46	1.28 ± 0.81	0.443	1.38 ± 0.76	1.08 ± 0.43	0.122	1.07 ± 0.59	1.33 ± 1.07	0.343
First RST	0.75 ± 0.62	0.71 ± 0.36	0.856	0.69 ± 0.40	0.89 ± 0.53	0.271	0.93 ± 0.51	0.67 ± 0.39	0.095	0.72 ± 0.48	0.73 ± 0.37	0.943
Max RST	0.40 ± 0.35	0.64 ± 0.43	0.122	0.59 ± 0.34	0.57 ± 0.41	0.905	0.66 ± 0.39	0.57 ± 0.34	0.571	0.61 ± 0.51	0.57 ± 0.32	0.790
Rectal Compliance	1.60 ± 2.46	1.07 ± 0.78	0.524	1.13 ± 1.37	1.51 ± 1.18	0.498	2.16 ± 2.47	0.94 ± 0.72	0.179	1.06 ± 0.86	1.31 ± 1.67	0.547

MRP, mean resting pressure; MSP, mean squeezing pressure; R, resting; S, squeezing; RST, rectal sensory threshold.

## Data Availability

The datasets during and/or analyzed during the current study are available from the corresponding author on reasonable request.

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
