# Peer review of "Effect of Biofeedback Therapy during Temporary Stoma Period in Rectal Cancer Patients: A Prospective Randomized Trial"

_jcm, 2021, doi:10.3390/jcm10215172_

Round 1
Reviewer 1 Report
I would like to congratulate authors with this very interesting study. In many patients after radical sphincter saving surgery for rectal cancer LARS is unavoidable, and significant LARS may affect overall quality of life significantly. There is a number of different modalities to treat LARS, but noe of the is a key to solve the problem. Therefor prevention of LARS can be not only a solution, at least to some patients, but as well energy saving option to treat this difficult condition.
Author Response
I would like to congratulate authors with this very interesting study. In many patients after radical sphincter saving surgery for rectal cancer LARS is unavoidable, and significant LARS may affect overall quality of life significantly. There is a number of different modalities to treat LARS, but noe of the is a key to solve the problem. Therefor prevention of LARS can be not only a solution, at least to some patients, but as well energy saving option to treat this difficult condition.
- Thank you for your nice review and comments. We are conducting a large-scale prospective study on this issue. So, we hope that we are looking forward to reporting positive results from our study shortly soon.
Reviewer 2 Report
Well written manuscript. I do not have specific comments
Author Response
Well written manuscript. I do not have specific comments
- Thank you for your nice review and comments.
Reviewer 3 Report
Dear Authors,
I would like to congratulate you with this fine study.
I have few queries.
INTRODUCTION
"SPS presumably translates into better quality of life compared to an abdominoperineal resection through the avoidance of a permanent stoma." - would insert a citation here: Kang SB, Cho JR, Jeong SY, Oh JH, Ahn S, Choi S, Kim DW, Lee BH, Youk EG, Park SC, Heo SC, Lee DS, Ryoo SB, Park JW, Park HC, Lee SM, Kang SI, Kim MH, Oh HK, Shin R, Kim MJ, Lee KH, Kim YH, Kim JS, Lee KW, Lee HS, Kim HJ, Park YS, Sohn DK, Park KJ; Seoul Colorectal Research Group (SECOG). Quality of life after sphincter preservation surgery or abdominoperineal resection for low rectal cancer (ASPIRE): A long-term prospective, multicentre, cohort study. Lancet Reg Health West Pac. 2020 Dec 28;6:100087.
Here "However, SPS may lead to significant bowel dysfunction, known as low anterior resection (LARS), which can adversely affect QoL" you can cite some - Dulskas A, Kavaliauskas P, Pilipavicius L, Jodinskas M, Mikalonis M, Samalavicius NE. Long-term bowel dysfunction following low anterior resection. Sci Rep. 2020 Jul 17;10(1):11882.
"However, there are no specific treatments for LARS and most of the treatment approaches are empirical and symptom-based, using existing therapies for fecal incontinence, fecal urgency, and rectal evacuatory disorders, including loperamide administration, anal plugs, biofeedback therapy (BFT), rectal irrigation, and neuromodulation with sacral or tibial nerve stimulation[9]" there is a fine review of possible treaments: Dulskas A, Smolskas E, Kildusiene I, Samalavicius NE. Treatment possibilities for low anterior resection syndrome: a review of the literature. Int J Colorectal Dis. 2018 Mar;33(3):251-260.
METHODS
No comments
RESULTS
Figure 2 quality must be improved
"However, there were marginally significant difference in gas incontinence (P=0.071), liquid stool incontinence (P=0.06), and lifestyle alteration (P=0.096) between BFT group and control group." The term "marginally significant" is not scientific. Actually, there was no statistical significance. This must be changed all over the text.
DISCUSSION
well written
Limitations - add single center; drop off of 21% of included patients is significant - comment on that.
Author Response
Dear Authors,
I would like to congratulate you with this fine study.
I have few queries.
INTRODUCTION
"SPS presumably translates into better quality of life compared to an abdominoperineal resection through the avoidance of a permanent stoma." - would insert a citation here: Kang SB, Cho JR, Jeong SY, Oh JH, Ahn S, Choi S, Kim DW, Lee BH, Youk EG, Park SC, Heo SC, Lee DS, Ryoo SB, Park JW, Park HC, Lee SM, Kang SI, Kim MH, Oh HK, Shin R, Kim MJ, Lee KH, Kim YH, Kim JS, Lee KW, Lee HS, Kim HJ, Park YS, Sohn DK, Park KJ; Seoul Colorectal Research Group (SECOG). Quality of life after sphincter preservation surgery or abdominoperineal resection for low rectal cancer (ASPIRE): A long-term prospective, multicentre, cohort study. Lancet Reg Health West Pac. 2020 Dec 28;6:100087.
Here "However, SPS may lead to significant bowel dysfunction, known as low anterior resection (LARS), which can adversely affect QoL" you can cite some - Dulskas A, Kavaliauskas P, Pilipavicius L, Jodinskas M, Mikalonis M, Samalavicius NE. Long-term bowel dysfunction following low anterior resection. Sci Rep. 2020 Jul 17;10(1):11882.
"However, there are no specific treatments for LARS and most of the treatment approaches are empirical and symptom-based, using existing therapies for fecal incontinence, fecal urgency, and rectal evacuatory disorders, including loperamide administration, anal plugs, biofeedback therapy (BFT), rectal irrigation, and neuromodulation with sacral or tibial nerve stimulation[9]" there is a fine review of possible treaments: Dulskas A, Smolskas E, Kildusiene I, Samalavicius NE. Treatment possibilities for low anterior resection syndrome: a review of the literature. Int J Colorectal Dis. 2018 Mar;33(3):251-260.
- Thank your for your nice and kind review. I added these references as your recommendation.
METHODS
No comments
RESULTS
Figure 2 quality must be improved
- I modified Figure 2.
"However, there were marginally significant difference in gas incontinence (P=0.071), liquid stool incontinence (P=0.06), and lifestyle alteration (P=0.096) between BFT group and control group." The term "marginally significant" is not scientific. Actually, there was no statistical significance. This must be changed all over the text.
- I changed some sentences in Results and Discussion section.
DISCUSSION
well written
Limitations - add single center; drop off of 21% of included patients is significant - comment on that.
- I added some sentences into Discussion section as your recommendation.
- Once again, I greatly appreciate your nice review and kind comments.